# Response of photosynthesis to different concentrations of heavy metals in *Davidia involucrata*

**Yan Yang**[1,2☯‡]*, **Liuqing Zhang**[1☯‡], **Xing Huang**[1], **Yiyang Zhou**[1], **Qiumei Quan**[1,2], **Yunxiang Li**[1,2], **Xiaohua Zhu**[1,2]*

1 College of Environment Science and Engineering, China West Normal University, Nanchong, China,
2 Institute of Environmental Sciences, China West Normal University, Nanchong, China

☯ These authors contributed equally to this work.
‡ YY and LZ are co-first authors.
* sister_yy@sina.cn (YY); 18990869488@163.com (XHZ)

**Data Availability Statement:** All relevant data are within the paper and its Supporting Information files.

**Funding:** This work was supported by the National Natural Science Foundation of China (Grant No.

## Abstract

Lead (Pb) and cadmium (Cd) are highly toxic and are widespread in agricultural soils, representing risks to plant and human health. In this study, *Davidia involucrata* was cultivated in soil with different concentrations of Pb and Cd and sampled after 90 days. We used ANOVA to analyse the photosynthesis of *D. involucrata* and the ability of Pb and Cd to enrich and migrate in roots, stems and leaves. Various results are described here. 1) Under individual and combined Pb and Cd stress, the accumulation factors in the roots were greater than 1, which was significantly greater than those in the stems and leaves ($P < 0.05$), and the translocation factors both were less than 1. The Pb and Cd enrichment ability of *D. involucrata* roots was significantly higher than that of stems and leaves, and the migration ability of the two heavy metals in *D. involucrata* was weak. 2) The Mg-dechelatase activities of chlorophyll degradation products increased under stress due to high concentrations of Pb and Cd. However, chlorophyllase activity was higher at relatively low concentrations of the two heavy metals ($P < 0.05$). δ-Aminolevulinic acid and porphobilinogen of chlorophyll synthesis products are easily converted to uroporphyrinogen III under low concentrations of Cd, which promotes the synthesis of chlorophyll. 3) The effect of Cd stress alone on the chlorophyll concentration was not significant. Under combined stress, concentrations of Pb and Cd in the range of 400~800 mg·kg⁻¹ and 5~20 mg·kg⁻¹ significantly promoted an increase in photosynthetic pigments ($P < 0.05$). 4) Inhibition of the net photosynthetic rate increased with increasing Pb and Cd concentrations under both individual and combined stress. In addition, the root of *D. involucrata* had a strong absorption and fixation effect on heavy metals, thereby reducing metal toxicity and improving the tolerance of *D. involucrata* to heavy metals.

31671688), the Meritocracy Research Funds of China West Normal University (Grant No. 17YC145), and the Fundamental Research Funds of China West Normal University (Grant No. 17E055).

**Competing interests:** The authors have declared that no competing interests exist.

**Abbreviations:** Total chl, Total chlorophyll; Chl-a, Chl-b and Car, Chlorophyll a, chlorophyll b and carotenoids; Chlase, Chlorophyllase; MDCase, Mg-dechelatase; δ-ALA, δ-aminolevulinic acid; PBG, Porphobilinogen; Urogen III, Uroporphyrinogen; BCF and TF, Accumulation factors and translocation factors.

# 1 Introduction

In recent decades, anthropogenic activities have accelerated the release of pollutants, especially heavy metals, into the environment, which has created potential hazards to ecosystems and human health [1,2]. Lead (Pb) and cadmium (Cd) are nondegradable, long-lived and exhibit strong toxicity in the soil [3]. They are highly toxic and pose a threat to plants and animals (including humans) by affecting their normal growth and health [4,5]. Heavy metals are absorbed mainly through the roots of plants and either remain there or are translocated to the shoots and into cells [6]. For most plant species, roots represent a barrier for metals. Therefore, the concentration of heavy metals in roots is usually higher than that of stems and leaves [7,8,9]. The toxicity of individual and combined Pb and Cd stress on photosynthesis is well documented [10,11]. Moreover, excessive Pb and Cd in the soil reduces the uptake of minerals and micronutrients by plants, interferes with plant water balance, inhibits stomatal opening, and decreases plant quality [12,13,14,15,16]. These stresses inhibit gas exchange and photosynthetic pigment biosynthesis because of the destruction of the chloroplast ultrastructure and the disassembly of thylakoids [17,18].

*Davidia involucrata* Baill., a member of Davidiaceae, is a rare and endangered tree species unique to China. This tree is a famous Tertiary relict plant and is referred to as a "living fossil". *D. involucrata* is highly valued for research, ornamental and medicinal purposes and has been widely introduced and cultivated in China. *D. involucrata* has been gradually introduced into foreign countries because of its ornamental value, which improves its economic value [19,20]. With increasing intensity of human activities and regional development, many pollutants released into the environment have caused a sharp decrease in naturally distributed areas and population numbers of *D. involucrata*, affecting the survival of introduced and cultivated plants [21,22]. Since the discovery of *D. involucrata* in 1869, numerous reports on this species have focused on its communities, botanical aspects, artificial propagation and cultivation techniques, population ecology, biological characterization, histochemistry, cytology, etc. [23,24,25]. However, few studies have investigated physiological and biochemical changes in response to Pb and Cd stress, including changes in photosynthesis [26,27]. Chlorophyll a, chlorophyll b and carotenoids constitute the main photosynthetic pigments. Chlorophyll a plays an important role in the oxygen production of photosynthetic plants, and chlorophyll b functions in absorbing blue light energy. Carotenoids regulate the growth and development of plants and the interactions between plants and the environment [28]. In addition, some substances related to the synthesis and decomposition of chlorophyll also indirectly affect photosynthetic function. Zhou et al. [29] reported that chlorophyllase (Chlase) and Mg-dechelatase (MDCase) can cause the decomposition of chlorophyll. In contrast, porphobilinogen (PBG), δ-aminolevulinic acid (δ-ALA) and uroporphyrinogen III (Urogen III) are closely related to chlorophyll production; heavy metals directly affect plants by modulating the activities of these enzymes, thus indirectly affecting the photosynthesis, growth, and yield of plants [14,30,31].

Several studies have investigated the physiological and biochemical effects of *D. involucrata* in response to heavy metal stress. We suspect that the photosynthesis of *D. involucrata* would be inhibited as the concentration of heavy metals increases. Moreover, the root system of *D. involucrata* may have a certain heavy metal-enrichment ability to resist stress. In this study, we examined the effects of individual and combined Pb and Cd stress on physiological and biochemical indexes of *D. involucrata* seedlings to determine the Pb and Cd tolerance mechanisms of *D. involucrata*. The results of this study could help to protect *D. involucrata* effectively and improve the survival rate of introduced and cultivated materials. Furthermore, this study could provide reference data, expanding the relevant information for research on *D. involucrata*.

## 2 Materials and methods

### 2.1 Plant material and growth conditions

The *D. involucrata* used in the present study was purchased from Shifang, Sichuan Province, China. Healthy and similar-sized seedlings were selected and sown in plastic pots at the experimental station of West Normal University in China. The soil used in the experiment was obtained from the experimental station; in terms of its physical and chemical properties, its pH was 7.76±0.07, and its total nitrogen (TN) and total phosphorus (TP) contents were 513.47 mg·kg$^{-1}$ and 472.5 mg·kg$^{-1}$, respectively. The background values of Pb and Cd in the soil were 5.71 mg·kg$^{-1}$ and 0.09 mg·kg$^{-1}$, respectively.

### 2.2 Experimental setup and management

In accordance with GB15618-1995 (Soil Environmental Quality Standards, GB15618-1995, China), the three levels of soil environmental quality standard values are Pb≤500 mg·kg$^{-1}$ and Cd≤1 mg·kg$^{-1}$. In China, the highest levels of Pb and Cd pollution can reach 1143 mg·kg$^{-1}$ and 228 mg·kg$^{-1}$, respectively [32]. We adopted an orthogonal experimental design method to establish 16 concentration gradients to simulate the effects of mild, moderate and severe pollution of heavy metals on the photosynthesis of *D. involucrata*. Pb(NO$_3$)$_2$ and CdCl$_2$·2.5H$_2$O were used to generate different concentrations of solutions, and *D. involucrata* was cultivated for 90 days. Pb and Cd stress were individually applied by adding 0, 200, 400, 600, 800, and 1000 mg·kg$^{-1}$ and 0, 1, 5, 10, 20, and 30 mg·kg$^{-1}$, respectively. Combined stress was applied by adding 0, 200 and 1, 400 and 5, 600 and 10, 800 and 20, and 1000 and 30 mg·kg$^{-1}$, with three replicates of all treatments.

### 2.3 Physiological measurements

**2.3.1 Photosynthetic pigments.** Five millilitres of acetone was added to 0.5 g of leaf tissue, which was incubated in darkness (4˚C for 72 h) until the colour completely disappeared from the leaves [33,34,35,36]. The samples were then centrifuged at 4000 g for 10 min at 4˚C, after which the supernatant was collected. The absorbance at 663 nm, 645 nm and 470 nm was measured by a UV755 spectrophotometer (China, Shanghai, UV755), and the concentrations of chlorophyll a, chlorophyll b and carotenoids were calculated according to the methods of Lichtenthaler et al. [33].

**2.3.2 Activities of chlorophyll synthesis and degradation products.** Two hundred milligrams of fresh leaf was weighed and thoroughly ground in liquid nitrogen. The tissue was then added to extraction solution (0.1 mmol PBS, pH 7.4) at a volumetric ratio of 1:9 (tissue: solution). The solution was subsequently incubated at 4˚C for 2 h and centrifuged at 3000 rpm for 10 min at 4˚C, after which the supernatant was used as the sample solution.

The activities of Chlase, MDCase, δ-aminolevulinic acid (δ-ALA), porphobilinogen (PBG) and uroporphyrinogen (Urogen Ⅲ) were measured by using a Chlase assay kit (LE-B044, 96T), MDCase assay kit (LE-B059, 96T), δ-ALA assay kit (LE-06543, 96T), PBG assay kit (LE-B255, 96T) and Urogen Ⅲ assay kit (LE-B254, 96T), respectively. The enzyme-linked immunosorbent assay kit (ELISA) produced by Hefei Laier Bioengineering Institute was implemented according to the manufacturer's instructions. The ELISA kit involves a one-step sandwich enzyme-linked immunosorbent assay with double antibodies. Extracts (10 μL) of the samples and sample diluents (40 μL) were added to precoated antibody micropores, after which 100 μL of horseradish peroxidase (HRP)-labelled antibodies was added to the micropore with the sample. The system was subsequently incubated in a constant-temperature box at 37˚C for 60 min. The micropore was cleaned with detergent, after which the substrates were

added. Afterward, fifty microliters of the substrate was added, followed by incubation at 37°C in the dark for 15 min. Finally, 50 μL of termination solution was added to each pore. The absorbance value [optical density (OD) value] of each pore was measured at a wavelength of 450 nm within 15 min by an enzyme-labeled instrument (Multiskan Go, THERMO, USA).

**2.3.3 Gas-exchange measurements.** The middle and upper function leaves (fully extended) of *D. involucrata* seedlings were selected to measure gas exchange. The net photosynthetic rate, stomatal conductance, intercellular $CO_2$ concentration, and transpiration rate were measured at the end of the experiment via an LI-6400 portable photosynthesis system (LI-COR, Lincoln, NE, USA) during the daytime—between 9:00 a.m. and 12:00 p.m.—under maximum daylight intensity [37].

## 2.4 Metal content analysis of plants

After 90 days, the whole plants were harvested, after which the roots and leaves were separated and dried at 65°C for 72 h to a constant weight to measure metal concentrations. The root, stem and leaf samples were digested in $HNO_3$-$HClO_4$, and the concentrations of Pb and Cd were determined via an atomic absorption spectrophotometer (AA-7000, Shimadzu, Japan) [6,38]. To explore the accumulation and transformation of Pb and Cd in the roots, stems and leaves, the bioaccumulation factor (BCF) and the translocation factor (TF) were calculated as follows [3]:

$$BCF = C_{roots}(mg \cdot kg^{-1}DMW)/C_{sub}(mg \cdot kg^{-1}DMW) \tag{1}$$

$$TF = C_{stems\ and\ leaves}(mg \cdot kg^{-1}DMW)/C_{roots}(mg \cdot kg^{-1}DMW) \tag{2}$$

where DMW, $C_{roots}$, $C_{stems\ and\ leaves}$ and $C_{sub}$ are dry matter weight and the metal concentrations in the plant roots, stems and leaves (mg·kg$^{-1}$ DMW) and soil (mg·kg$^{-1}$ DMW), respectively. The BCF and TF can be used to characterize the ability of plants to accumulate and translocate heavy metals, respectively. High BCF values and TF values indicate that the ability of plants to accumulate and translocate heavy metals to the aboveground plant parts is strong.

## 2.5 Statistical analyses

The experimental results of this study are presented as the mean of three replicates. Differences among treatments were analysed by one-way analysis of variance (ANOVA), and the significance of interactions between Pb and Cd was analysed by two-way ANOVA. The least significant difference (Tukey's test) was applied to determine the significance between different treatments, and the critical value for statistical significance was $P < 0.05$. All statistical analyses were carried out using SPSS 23.0 (SPSS, Chicago, USA).

## 3 Results

### 3.1 Attributes of photosynthetic pigments

Fig 1 shows the variation of photosynthetic pigment concentrations. When the Pb concentration was 400 mg·kg$^{-1}$, the total chlorophyll, chlorophyll a and chlorophyll b contents were the greatest, and these levels were significantly greater than those under the control treatment. When the Pb concentration was 1000 mg·kg$^{-1}$, the concentration of carotenoids was significantly greater ($P<0.05$) than that under the other treatments. There was no significant difference in chlorophyll a or total chlorophyll after treatment with different concentrations of Cd. The level of chlorophyll b under the control treatment was significantly greater than that when 1 mg·kg$^{-1}$ and 10 mg·kg$^{-1}$ Cd were added ($P<0.05$). The concentration of carotenoid was

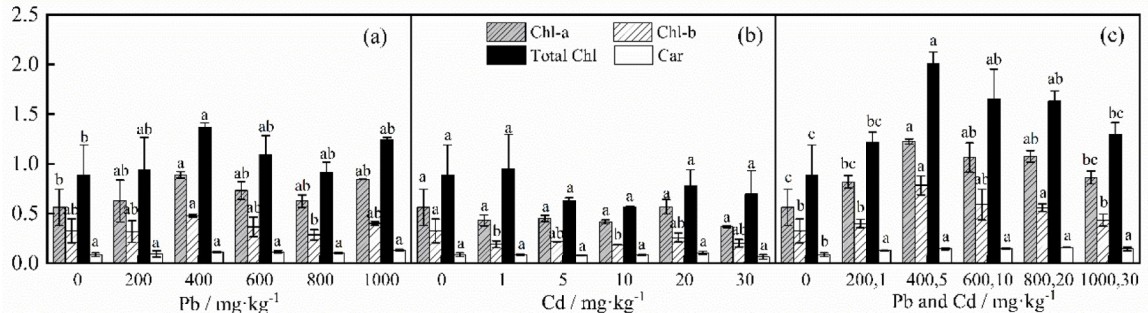

**Fig 1. Analysis of the differences in photosynthetic pigments of *D. involucrata* under different concentrations of Pb and Cd.**

lowest ($0.062 \pm 0.024$ mg·kg$^{-1}$) at 30 mg·kg$^{-1}$ Cd. When the Cd concentration was 20 mg·kg$^{-1}$, the concentration of carotenoids was significantly greater than that at 30 mg·kg$^{-1}$. When Pb and Cd were added concurrently, the contents of the four photosynthetic pigments in the control treatment were significantly lower than those under the other treatments. There was no significant difference in the photosynthetic pigments at 400+5–800+20 mg·kg$^{-1}$.

## 3.2 Characteristics of chlorophyll synthesis and degradation products

**3.2.1 Effects of Pb and Cd on chlorophyll degradation products.** Chlase and MDCase can promote the decomposition of total chlorophyll and indirectly affect the photosynthetic capability of plants. As shown in Fig 2A, Chlase responded similarly to individual Cd and combined stress and exhibited maximum activity in the control group. When Pb stress alone was 400 mg·kg$^{-1}$, Chlase activity was significantly greater than that under the other treatments ($P<0.05$). Chlase activity was significantly lower under 1 mg·kg$^{-1}$ added Cd than under the control treatment, and Chlase activity was highest under the combined stress of 400+5 mg·kg$^{-1}$. Fig 2B shows that when the Pb concentration reached 200 kg·mg$^{-1}$ and 1000 mg·kg$^{-1}$, the activity of MDCase was significantly greater than that under other treatments. However, compared with the control treatment, MDCase activity in response to 400–800 mg·kg$^{-1}$ added Pb was not significantly affected. MDCase exhibited the highest level of activity when the Cd concentration was 30 mg·kg$^{-1}$, and its activity was significantly greater than that under other treatments. The same effect on MDCase activity was observed under 1–20 mg·kg$^{-1}$ added Cd. The response of MDCase to combined stress was similar to its response to Cd stress alone.

**3.2.2 Effects of Pb and Cd on chlorophyll synthesis products.** The responses of δ-ALA, PBG and Urogen III to individual and combined Pb and Cd stress exhibited different patterns [Fig 3A–3C]. When the concentrations of Pb stress alone exceeded 400 mg·kg$^{-1}$, the δ-ALA content gradually increased, and the δ-δ-ALA content was significantly greater under 400 mg·kg$^{-1}$ added Pb than under the control treatment. The δ-ALA content reached the highest level under 20 mg·kg$^{-1}$ added Cd and significantly differed from that under the control treatment. When the Cd concentration reached 1–30 mg·kg$^{-1}$, the δ-ALA content was significantly lower than that under the control treatment. When the combined stress was 30+1000 mg·kg$^{-1}$, the δ-ALA content increased, but there was no significant difference in the content between the other treatments and the control treatment. The effect of individual Pb and Cd stress on δ-ALA was greater than the effect of combined stress. PBG initially decreased but then increased under Pb stress alone; its lowest level was detected under 400 mg·kg$^{-1}$ added Pb. When the Cd concentration reached 20 mg·kg$^{-1}$, the PBG content was greatest and did not significantly differ from that under the control treatment. Under Cd stress alone, the content of PBG under the control treatment was significantly greater than that under the other treatments ($P<0.05$).

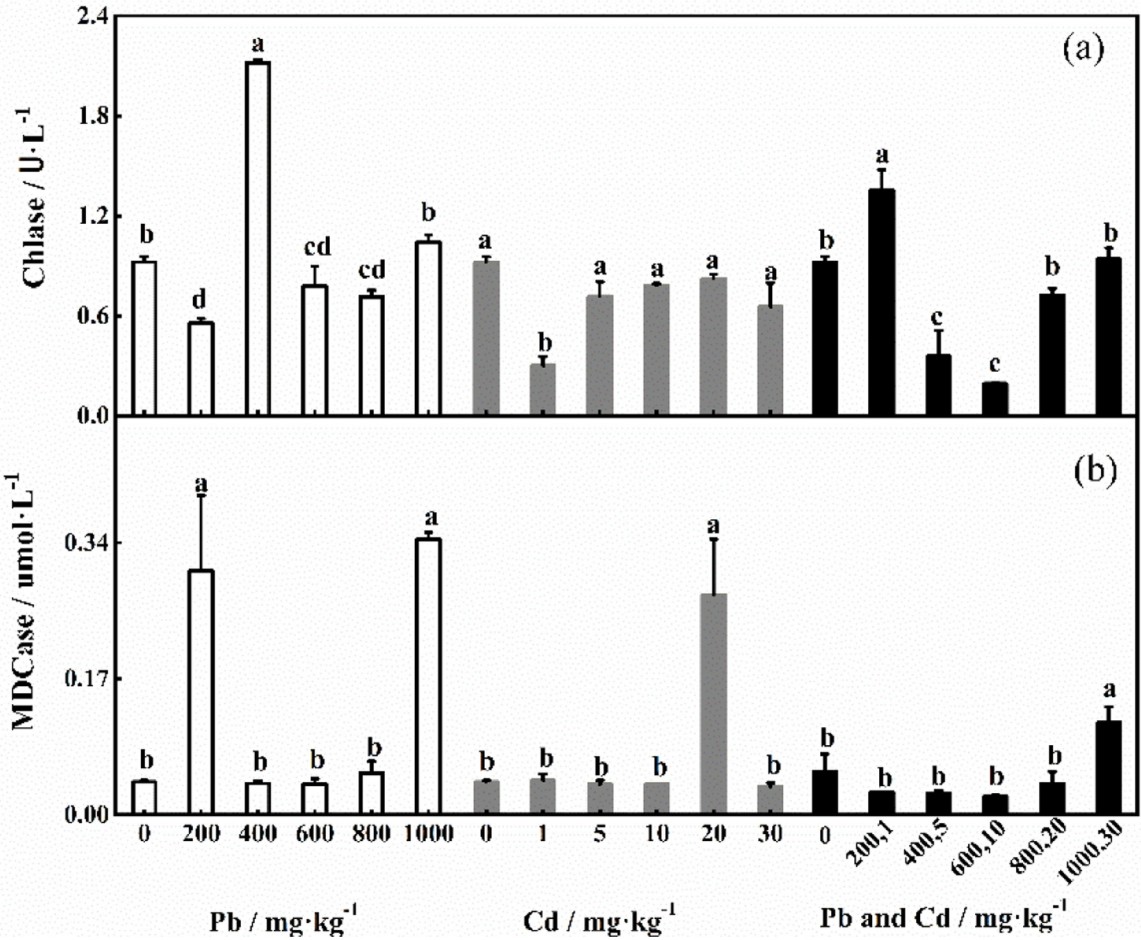

**Fig 2. Effects of different concentrations of Pb and Cd on the activities of chlorophyllase and Mg-dechelatase.**

The content of Urogen III was highest under 200 mg·kg$^{-1}$ added Pb and significantly lower under 800 mg·kg$^{-1}$ than that under the other treatments ($P<0.05$). Fig 3C shows that the Urogen III content initially increased, decreased when the Cd concentration was 1 mg·kg$^{-1}$ and 5 mg·kg$^{-1}$ and then peaked under 5 mg·kg$^{-1}$. The content of Urogen III was significantly different at different concentrations of Cd ($P<0.05$).

### 3.3 Effects of Pb and Cd on gas-exchange parameters

The net photosynthetic rate significantly increased under 200 mg·kg$^{-1}$ added Pb and was significantly different from that under the control treatment [Fig 4A]. The net photosynthetic rate gradually decreased when the Pb concentration exceeded 200 mg·kg$^{-1}$. The variation in the stomatal conductance and transpiration rate was similar to that in the net photosynthetic rate [Fig 4B and 4D]. The intercellular $CO_2$ concentration reached the lowest level under 600 mg·kg$^{-1}$ added Pb but was significantly greater than that under the treatments with 800 mg·kg$^{-1}$ and 1000 mg·kg$^{-1}$. As shown in Fig 5A, the net photosynthetic rate in the control treatment was significantly greater ($P<0.01$) than that under Cd stress alone. When the Cd concentration was 20 mg·kg$^{-1}$, the stomatal conductance was not significantly different from that under the control treatment, and the trend was similar to that of Pn [Fig 5B]. The intercellular $CO_2$ concentration was significantly greater at 1–30 mg·kg$^{-1}$ Cd concentrations than that under the

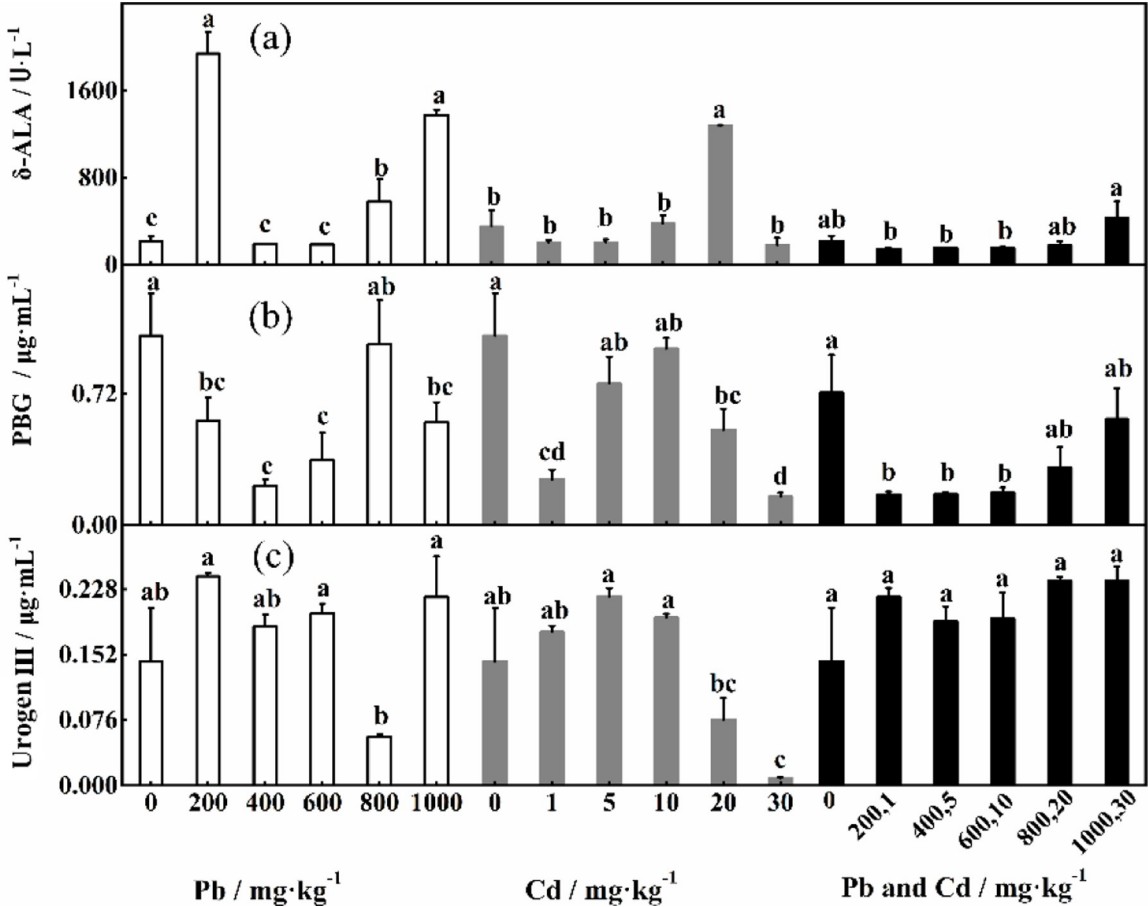

**Fig 3. Effects of different concentrations of Pb and Cd on the contents of δ-aminolevulinic acid porphobilinogen, and uroporphyrinogen III.**

control treatment, and the intercellular $CO_2$ concentration peaked under 10 mg·kg$^{-1}$ Cd [Fig 5C]. The transpiration rate was significantly greater under 1 mg·kg$^{-1}$ and 5 mg·kg$^{-1}$ added Cd than under the control treatment and reached the lowest level at 10 mg·kg$^{-1}$ Cd. The trends of the net photosynthetic rate and stomatal conductance were similar under combined stress, that is, an initial decrease followed by an increase with increasing Pb and Cd. The net photosynthetic rate was greatest under the control treatment. The transpiration rate also increased with increasing stomatal conductance (Fig 6).

## 3.4 Accumulation and distribution characteristics of Pb and Cd

As shown in Fig 7, the BCF values of the roots were greater than those of the stems and leaves. The BCF values of the roots decreased significantly with increasing Pb and Cd concentrations. These results indicated that only a small portion of Pb and Cd is translocated to the stems and leaves. Regardless of the presence of individual or combined stress, the BCF values under the control treatment were significantly greater than those under other treatments, and the values in response to combined stress were lower than those in response to individual stresses ($P<0.05$). Combined stress could reduce the ability of Pb and Cd to be translocated from the underground plant parts to the aerial parts. The BCF values of the stems and leaves did not

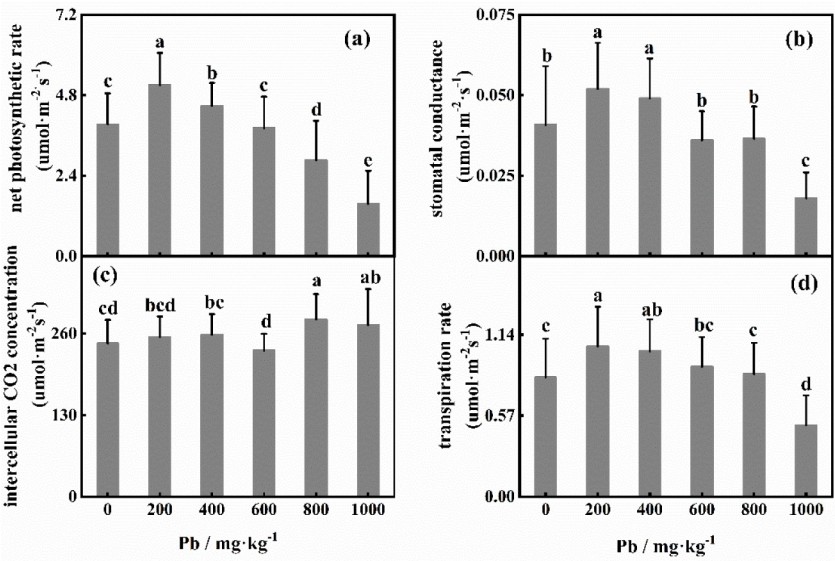

Fig. 4 Effects of different concentrations of Pb on gas-exchange parameters

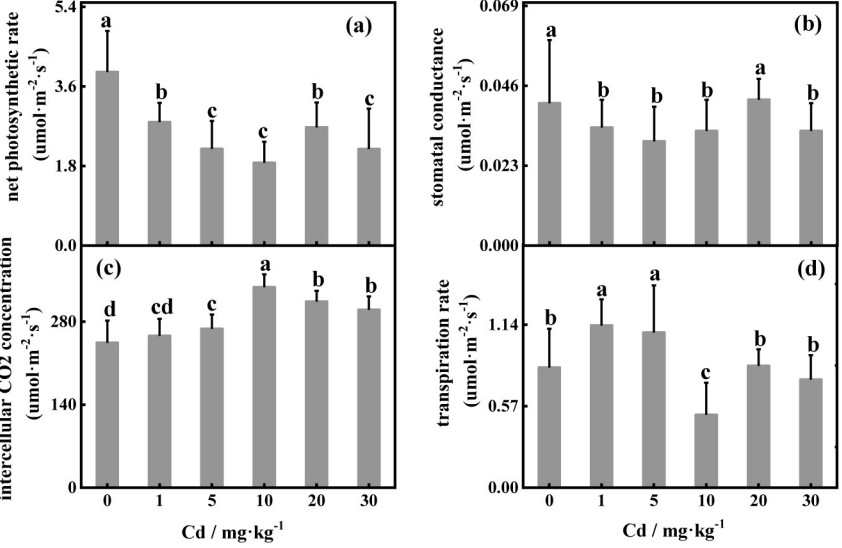

**Fig 4. Effects of different concentrations of Pb on gas-exchange parameters.**

significantly differ in response to the increased Pb and Cd concentrations under individual and combined stress.

On the other hand, the TF values strongly reflected the translocation of heavy metals in the plants. The TF values decreased with increasing heavy metal concentrations (Table 1). The TF values were greatest when the Pb concentration was 400 mg·kg$^{-1}$, but these values did not significantly differ from those under the control treatment. The TF values were significantly greater under the control treatment than under treatments with 200–1000 mg·kg$^{-1}$ added Pb. Under Cd stress alone, the TF values under the control treatment were greatest and were not significantly different from those under the 1 mg·kg$^{-1}$ Cd treatment. When the Cd concentration exceeded 5 mg·kg$^{-1}$, the TF value was not significantly different from that at 10–30 mg·kg$^{-1}$ Cd.

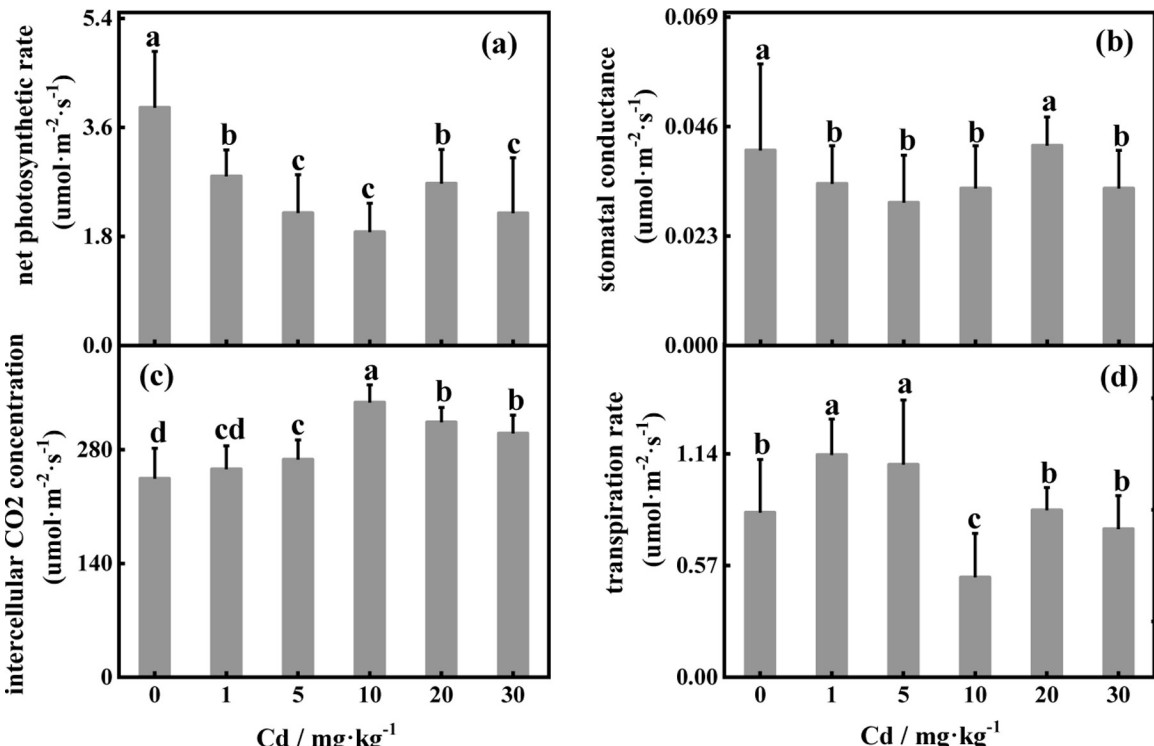

**Fig 5. Effects of different concentrations of Cd on gas-exchange parameters.**

The changes in the TF values of individual Pb and Cd stress were similar to those under combined stress.

## 4 Discussion

### 4.1 Response of photosynthetic pigments to Pb and Cd

Chlorophyll and carotenoids, which are photosynthetic pigments, are important substances in plants for the conversion of solar energy into chemical energy. These pigments guarantee that plants are able to synthesize their own substances [39]. With an increase in the concentration of single Pb and combined stress, Chl-a, Chl-b and total chlorophyll showed increasing trends, and the concentrations were higher than those of the control group. These results may be due to the strong tolerance of *D. involucrata* to Pb and Cd, which may be attributable to the root system of *D. involucrata*, which has strong adsorption and retention of heavy metals and reduces the toxicity of heavy metals to the leaves. Another explanation may be the chelation of phytochelatins to heavy metals, thereby reducing toxicity [40,41]. Similar to the findings of Figlioli et al. [42], this phenomenon was attributable to the binding of Pb and Cd, which reduced the toxicity of their individual actions. The concentrations of Chl-a and Chl-b were lower than those of the control group under Cd treatment, except that the Cd concentration was 20 kg·mg$^{-1}$, but there was no significant difference in Chl-a and Chl-b concentrations with increased concentrations of Cd, indicating Cd strongly inhibits chlorophyll. However, we also found that there was no significant change in the concentration of carotenoids under either single or combined stress, indicating carotenoids are not sensitive to Pb and Cd stress [43].

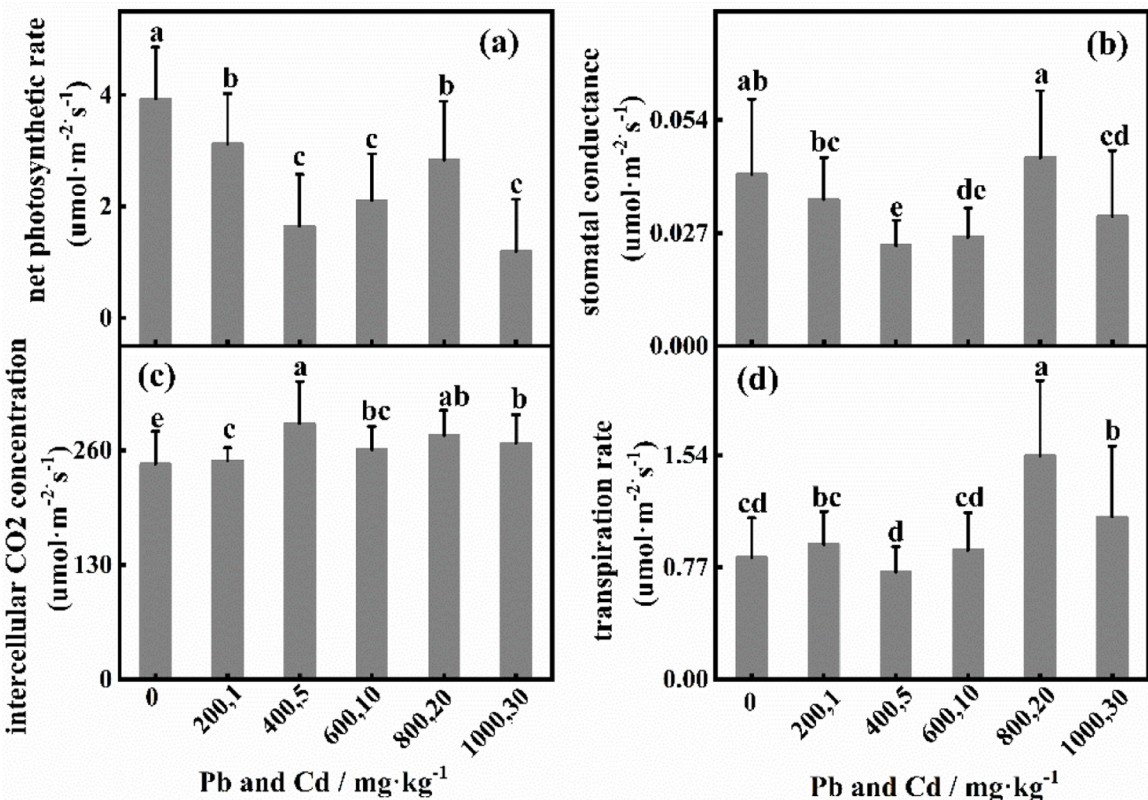

**Fig 6. Effects of different concentrations of Pb and Cd on gas-exchange parameters.**

## 4.2 Response of chlorophyll synthesis and degradation products

Chlase and MDCase are involved mainly in the degradation of chlorophyll, and these products can indirectly reflect changes in chlorophyll [7]. Chlase and MDCase play key roles in the first and second steps of the chlorophyll decomposition process, respectively, showing high activity at a Pb concentration of 400 mg·kg$^{-1}$, Cd concentrations of 10 mg·kg$^{-1}$ and 20 mg·kg$^{-1}$ and combined stress concentrations of 200 + 1 mg·kg$^{-1}$. Under stress due to high concentrations of Pb and Cd (1000 mg·kg$^{-1}$ and 30 mg·kg$^{-1}$), the activity of demerged chelatase, which plays a role in the second step of decomposition, was higher than that in the control group. This indicated that the degree of chlorophyll decomposition did not increase significantly under the stress of low heavy metal concentrations. Therefore, the increase in the chlorophyll concentration is related to the lower degree of chlorophyll decomposition [44]. This also may be related to the enhancement of other resistance mechanisms in plants. Yuan et al. [45] reported that antioxidants have a certain protective effect on plants, improving their ability to resist stress. Chlorophyll content under combined stress was greater than that under individual stresses, which further verified that the tolerance of *D. involucrata* increased in the combined stress environment. This effect may be due to the inhibition of Chlase and MDCase activities by combined stress or the physical and chemical effects of Pb and Cd on the soil. Some Pb and Cd may have been retained in the soil to reduce the stress effect on *D. involucrata* [46].

δ-ALA, PBG, and Urogen III play fundamental roles in photosynthesis, as they are involved mainly in the biosynthesis of chlorophyll. δ-ALA is converted to PBG by δ-aminolevulinic acid dehydratase, and then porphobilinogen is further converted to Urogen III by porphobilinogen deaminase [47]. δ-ALA, a key enzyme involved in the first step of chlorophyll synthesis,

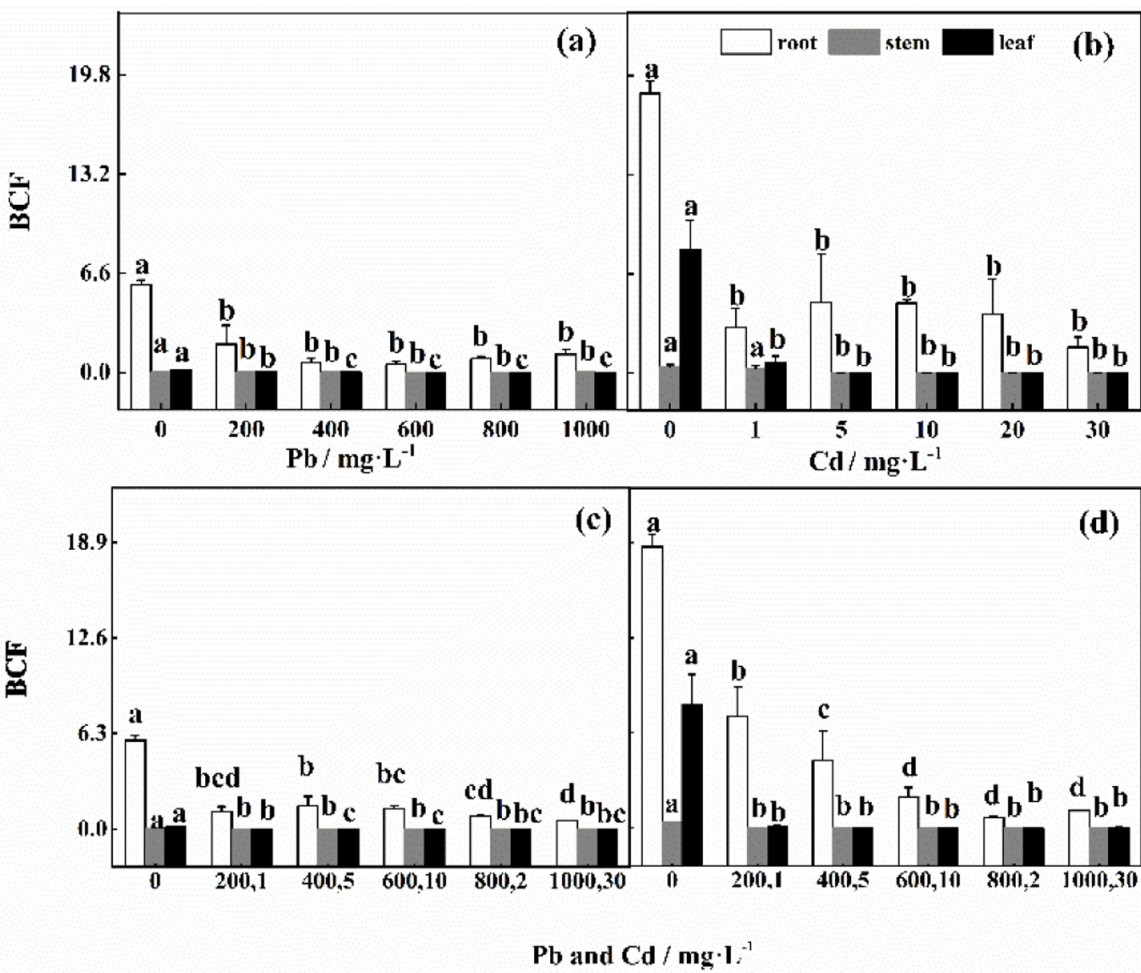

**Fig 7.** (a), (b) BCFs of Pb and BCFs of Cd in different plant parts under individual stresses. (c), (d) BCFs of Pb and BCFs of Cd in different plant parts under combined stress.

**Table 1. TF of Pb and Cd of *D. involucrata* under individual and combined stress[1].**

| Treatment level(mg·kg⁻¹) | | Single stress | Compound stress |
|---|---|---|---|
| Pb | CK | $0.033 \pm 0.004^{ab}$ | $0.033 \pm 0.00^{4a}$ |
| | 200 | $0.033 \pm 0.009^{ab}$ | $0.005 \pm 0.003^{b}$ |
| | 400 | $0.053 \pm 0.030^{a}$ | $0.002 \pm 0.001^{b}$ |
| | 600 | $0.025 \pm 0.015^{b}$ | $0.003 \pm 0.001^{b}$ |
| | 800 | $0.015 \pm 0.005^{b}$ | $0.003 \pm 0.000^{b}$ |
| | 1000 | $0.017 \pm 0.007^{b}$ | $0.003 \pm 0.001^{b}$ |
| Cd | CK | $0.461 \pm 0.089^{a}$ | $0.461 \pm 0.089^{a}$ |
| | 1 | $0.429 \pm 0.354^{a}$ | $0.031 \pm 0.003^{b}$ |
| | 5 | $0.008 \pm 0.003^{b}$ | $0.007 \pm 0.005^{b}$ |
| | 10 | $0.004 \pm 0.001^{b}$ | $0.006 \pm 0.006^{b}$ |
| | 20 | $0.004 \pm 0.003^{b}$ | $0.011 \pm 0.002^{b}$ |
| | 30 | $0.003 \pm 0.002^{b}$ | $0.040 \pm 0.014^{b}$ |

1) Different letters within the same column indicate significance at 5%

increased with increasing concentrations of individual and combined stress. However, with increases in single Pb and Cd concentrations, the concentration of porphobilinogen decreased and was lower than that of the control group, indicating that although δ-ALA increased, the synthesis of PBG was still strongly inhibited by heavy metals. Compound stress also inhibited the synthesis of PBG, which may be more sensitive to the toxicity of heavy metals [48]. The Urogen III content exhibited the "low promotion and high inhibition" phenomenon with an increase in individual Cd concentrations. Under single Pb treatment, the concentration of Urogen III was higher than that of the control group except for 800 mg·kg$^{-1}$. The concentration of Urogen III increased and was higher than that in the control group under combined stress, which further confirmed that the chlorophyll concentration was less inhibited by heavy metals. This effect may ensure that the inhibition of photosynthesis is reduced when the plant is exposed to environmental stress [49,50]. However, these findings differ from those of Li et al. [51], which may result from differences between species, or the enzymes involved may be highly resistant to heavy metal stress [52].

## 4.3 Response of gas-exchange parameters to Pb and Cd

The net photosynthetic rate decreased significantly with increasing Pb concentration and peaked at Pb of 200 mg·kg$^{-1}$, indicating that *D. involucrata* photosynthesis was promoted at low concentrations of Pb. This result can be verified by the same variation observed in the stomatal conductance and net photosynthetic rate. Xu et al. [53] also reported the same findings. Intercellular $CO_2$ concentration can be used to determine whether this effect is stomatal or nonstomatal. In the present study, the intercellular $CO_2$ concentration increased, and the transpiration rate increased first and then decreased with increasing Cd concentrations, which indicated that stomatal limitation is not the main factor affecting the photosynthesis of *D. involucrate* [53,54]. Under Cd stress alone, the net photosynthetic rate was lower than that under the control treatment and reached the lowest level at 10 mg·kg$^{-1}$ added Cd, indicating that, compared with Pb, Cd had a stronger inhibitory effect on photosynthesis. The intercellular $CO_2$ concentration exhibited a similar trend in response to individual Pb and Cd stresses, which indicates that Pb and Cd have similar effects on the gas-exchange parameters of *D. involucrata* photosynthesis. When the combined stress was at the highest concentration, the stomatal conductance, intercellular $CO_2$ concentration and transpiration rate did not decrease to their minimum. This may have occurred because the combination of Pb and Cd stress may have reduced the toxicity of the metals [7].

## 4.4 Accumulation and translocation of Pb and Cd in *D. involucrata*

The variation in heavy metal toxicity depends on plant species, type of metal and concentration, and soil composition [55]. Wu et al. [10] reported that the accumulation capacity of rape under Cd stress was greater than that under Pb stress for both individual and combined stress and that the toxic effect of Cd was greater than that of Pb. In addition, Rai et al. [39] reported that magnesium (Mg) within the chlorophyll ring is easily replaced by Cd. These results are consistent with those of our study. This phenomenon may have occurred because Cd is more easily absorbed by plants, and Pb is more likely to form sediment in the soil [8]. This consistency can also be confirmed by the high contents of δ-ALA and PBG under high concentrations of Pb and Cd. The accumulation capability of the stems and leaves of *D. involucrata* under combined stress was lower than that under individual stresses. These findings indicated that the interaction between Pb and Cd reduced their toxicity, and the strong isolation of the roots further reduced the toxicity. Under single and combined stress, the TF was less than 1, further confirming the results. These results are similar to those of Liu et al. [56]. However, in

our study, there was no obvious accumulation in various parts of the *D. involucrata* plants when the Pb concentration was high. This discrepancy may result from a combination of physical and chemical processes, such as soil uptake, root interception and resistance mechanisms [26]. These need further study.

## 5. Conclusion

The photosynthesis characteristics of *D. involucrata* in response to heavy metal stress and the Pb and Cd accumulation and translocation ability in different tissues were comprehensively reported for the first time. Photosynthetic pigments were slightly inhibited by Pb and Cd, and the synthesis of chlorophyll by *D. involucrata* was less affected by high concentrations of Pb and Cd. Chlorophyll synthesis products increased with increasing concentrations of Pb and Cd, and at the same time, the degradation products decreased. Gas-exchange parameters were more sensitive to heavy metal stress, and the net photosynthetic rate decreased with increased heavy metal concentrations. Almost all Pb and Cd was retained in the roots of *D. involucrata*, which reduced their toxicity in the stems and leaves. Moreover, there was low levels of translocation of these metals to the stems and leaves. These findings provide new perspectives on the photosynthesis tolerance of *D. involucrata* to environmental stress. Adding heavy metal fixatives to the soil could reduce the accumulation of heavy metals in the roots of *D. involucrata*, protecting the root tissues and ensuring normal growth of *D. involucrata*.

## Supporting information

**S1 Table. The mean and standard deviation of photosynthetic pigments of *D. involucrata* under different concentrations of Pb and Cd.**
(DOCX)

**S2 Table. The mean and standard deviation of chlorophyll synthesis and degradation products of *D. involucrata* under different concentrations of Pb and Cd.**
(DOCX)

**S3 Table. The mean and standard deviation of gas-exchange parameters of *D. involucrata* under different concentrations of Pb and Cd.**
(DOCX)

**S4 Table. The mean and standard deviation of accumulation factors(BFC) of *D. involucrata* under different concentrations of Pb and Cd.**
(DOCX)

## Author Contributions

**Conceptualization:** Yan Yang, Qiumei Quan.

**Data curation:** Liuqing Zhang, Xing Huang.

**Formal analysis:** Liuqing Zhang, Xing Huang.

**Investigation:** Yiyang Zhou.

**Project administration:** Yan Yang, Xiaohua Zhu.

**Software:** Qiumei Quan, Yunxiang Li, Xiaohua Zhu.

**Validation:** Yiyang Zhou.

**Visualization:** Qiumei Quan, Yunxiang Li.

**Writing – original draft:** Yan Yang, Liuqing Zhang.

**Writing – review & editing:** Yan Yang, Xiaohua Zhu.

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
