## [Decision Letter · Decision Letter 0]

1 Nov 2019

PONE-D-19-28304

Response of photosynthesis to different concentrations of heavy metals in Davidia involucrata

PLOS ONE

Dear Dr. Yang,

Thank you for submitting your manuscript to PLOS ONE. After careful consideration, we feel that it has merit but does not fully meet PLOS ONE’s publication criteria as it currently stands. Therefore, we invite you to submit a revised version of the manuscript that addresses the points raised during the review process.

The reviewers have raised some serious issues regarding the statistical analyses of the data as they are not appropriate and sound. The author should check carefully about the statistical results of all the data.  Additionally, the manuscript was written in a very casual manner, and the language of the manuscript need to be polished by a native speaker or an expert who write English well.

We would appreciate receiving your revised manuscript by 15th. Nov. 2019. To enhance the reproducibility of your results, we recommend that if applicable you deposit your laboratory protocols in protocols.io, where a protocol can be assigned its own identifier (DOI) such that it can be cited independently in the future. For instructions see: http://journals.plos.org/plosone/s/submission-guidelines#loc-laboratory-protocols

We look forward to receiving your revised manuscript.

Kind regards,

Mayank Gururani

Academic Editor

PLOS ONE

Journal Requirements:

Reviewers' comments:

Reviewer's Responses to Questions

**Comments to the Author**

1. Is the manuscript technically sound, and do the data support the conclusions?

Reviewer #1: Yes

Reviewer #2: Yes

Reviewer #3: No

2. Has the statistical analysis been performed appropriately and rigorously? 

Reviewer #1: No

Reviewer #2: No

Reviewer #3: Yes

3. Have the authors made all data underlying the findings in their manuscript fully available?

Reviewer #1: Yes

Reviewer #2: Yes

Reviewer #3: Yes

4. Is the manuscript presented in an intelligible fashion and written in standard English?

Reviewer #1: No

Reviewer #2: Yes

Reviewer #3: Yes

5. Review Comments to the Author

Reviewer #1: Overall, the manuscript was written in a very casual manner, and the language of the manuscript need to be polished by a native speaker or an expert who wrtie English well.

Lines 10-11: “We used one-way ANOVA to analyze the photosynthesis, accumulation and translocation in D. involucrate”. What accumulation and translocation? Need to be specific.

What do you mean “accumulation ability” in line 13, BCF or TF?

Line 15: roots of plant always have higher enrichment for HMs than stems and leaves.

Line 17: “increased under high concentrations stresses.”, high conc. of what stress? Need to be specified.

Line 18, “tow” should be “two”

Line 22: “Pb and Cd in the range of 400+5~800+20 mg kg-1”, what is the symbol “+” stands for?

Line 25: “The photosynthesis of D. involucrata has strong tolerance to higher concentrations of heavy metals.”, what do the authors mean "photosynthesis has strong tolerance"?

Lines 33-34: ). “Lead (Pb) and cadmium (Cd) are nondegradable, are easily translocated, and are long-lived in the soil”, The translocation ability of these two metals varied with plant species, can be easier in some plants but harder in some others. To my point of view, Pb is always hard to be translocated in most plants.

Lines 38-39: “The absorption capacity for heavy metals of different plants parts is as follows: roots > stems > leaves (Arena et al., 2017; zhu et al., 2007; Keller et al., 2003). ”. The absorption capacity of HMs in plants depends on the plant species and metal types. not for all the plants, hyperaccumulator sometimes accumulate more HMs in the aboveground parts.

Lines 96-97: The conc. of Pb and Cd used in the present study are too high, the author should state the reason in the manuscript why they choose such high levels of HMs and prove the levels used were environmental realistic.

Line 100: Do you have any reference to support the using of acetone to extract the leaf chlorophylls?

Line 105: which reference exactly is the one that your calculation based on?

Lines 129-134: What is the determination limit? Did you use any certified reference materials to check the recovery rate of your digestion process, and what is the recovery rate of these metals?

For the calculation of BCF and TF, I believe there are more suitable reference can be cited other than Wu et al. (2012)

Line 151: “were greatest” should be “were the greatest”

Figure 1: The abbreviations, Chlase and MDCase in Figure 1, are not defined previously.

The color of different columns in Fig.1: please consider to change the columns for Pb treatment to white. Same problems also shown for Fig.2 and fig.6.

For the data shown in Figures 1-5: the data of the different treatment does not seem to be significantly different from each other. You mentioned a Duncan test at 5% probabilities was used for comparison. Duncan's test is not a good choice when it comes to comparing treatment means. Authors should use an appropriate statistical test for this purpose and modify the content of the manuscript.

Some data in Figs. 4 do not seem to be significantly different. Take Figure 4a as an example, I don’t believe the data B4 and B5 are significantly different. The author should double-check the statistical results of the data in all the figures.

Table 2: Define “transfer coefficients”. Please include the calculation details in the materials and methods part.

Table 1 and Table 2: Using superscript lower-case letters to indicate the significant differences.

Line 342: “This study is the first to report the responses of..” the first what? need to rephrase.

Line 345: “It was found that the tolerance of D. involucrata to high concentration of Pb and Cd was restored.”, what do you mean restored? restored by what? Need to specify.

Line 348: “The photosynthetic pigments were relatively little affected by the heavy metals.”， please rephrase this sentence.

Reviewer #2: This is a nice paper reporting results about the "Response of photosynthesis to different concentrations of heavy metals in Davidia involucrata" - I would try to specify it more clearly and loudly in both the abstract and the conclusion, which are pretty weak now, could be much more catchy.

The study seems to be run using sounds laboratory techniques, but not robust statistical methods and data interpretation. Particularly, since the Authors are testing 2 different heavy metals (Pb and Cd) a 2-way ANOVA is needed.

I like this manuscript because so straightforward and a better statistical analysis of the results would increase the value. Also, since this plants is endemic from China, maybe adding a photo of the species would be a good idea.

Reviewer #3: The study deals with important topic of heavy metal effects in plants. The authors tried to address the effects of different concentrations of Pb and Cd on photosynthesis in a very specific wood plant Davidia involucrata, an endangered woody species. The authors focused on allocation of heavy metals in plants and effects of different concentrations on chlorophyll content, photosynthesis, stomatal activity as well as selected biochemical analyses. Overall, the manuscript is written well and presented clearly. However, I have some serious doubts related to some methodical aspects and interpretation of the data. Especially, the lead (Pb) was applied in form of nitrogen containing molecule Pb(NO)2. Despite the content of nitrogen in this molecule is not very high (app. 10%), considering a very high doses applied, the plants got significant amount of nitrogen, which could lead to significant fertilizing effects. This well explains increase of chlorophyll content or trend of photosynthetic characteristics observed in this study. The effect of additional nitrogen was not considered in methodic nor in interpretation of the data. Hence, there is a high risk that the recent version of the study contains significant artifacts and, therefore, it is not suitable for publication in present form. I strongly encourage the authors re-consider the results and use only the part in which there is no doubt regarding the methods.

In my point of view, the manuscript is not appropriate for publishing and should be rejected.

6. PLOS authors have the option to publish the peer review history of their article (what does this mean?). If published, this will include your full peer review and any attached files.

Reviewer #1: No

Reviewer #2: Yes: Lorenzo Rossi

Reviewer #3: No

---

## [Author Response · Author response to Decision Letter 0]

6 Jan 2020

Dear《PLOS ONE》editors:

We thank the editors and reviewers for their careful review of this article and for providing valuable comments and suggestions. After carefully reading the opinions of the two reviewers, I hope you will be satisfied with the following replies. If the revisions are not acceptable, we will revise and supplement the manuscript again according to the reviewers’ opinions.

Reviewer #1: Overall, the manuscript was written in a very casual manner, and the language of the manuscript need to be polished by a native speaker or an expert who wrtie English well.

1. Reply:

 We thank the reviewer for the valuable comments. We once again asked the professionals to refine the overall language of the manuscript. 

(1) Line 8. “a risk” has modified to “risks”.

(2) Line 8~9. “Davidia involucrata were cultivated in soil with different concentrations of Pb and Cd and sampled after 90 days” has modified to “Davidia involucrata was cultivated in soil with different concentrations of Pb and Cd and sampled after 90 days”

(3) Line 9~11. “We used ANOVA to analyze the photosynthesis of D. involucrata” has modified to “We used ANOVA to analyse the photosynthesis of D. involucrata”.

(4) Line 11. “The various results” has modified to “Various results”.

(5) Line 13. “than that” has modified to “than those”.

(6) Line 14~15. “It has been found that the enrichment ability of roots of D. nvolucrata to Pb and Cd is significantly higher than that of stems and leaves, and the migration ability of two heavy metals in D. involucrata is weak” has modified to “The Pb and Cd enrichment ability of D. involucrata roots was significantly higher than that of stems and leaves, and the migration ability of the two heavy metals in D. involucrata was weak.”

(7) Line 16~17. “The Mg-dechelatase activities of chlorophyll degradation products increased under high concentrations stresses of Pb and Cd” has modified to “The Mg-dechelatase (MDCase) activities of chlorophyll degradation products increased under stress due to high concentrations of Pb and Cd”

(8) Line 18. We have added the word “the”.

(9) Line 23. “The inhibition” has modified to “Inhibition”.

(10) Line 24. “combined stresses” has modified to “combined stress”.

(11) Line 46. “ the Davidiaceae” has modified to “Davidiaceae”.

(12) Line 68. “by affecting” has midified to “by modulating”.

(13) Line 74. “we have examined the effects” has modified to “we examined the effects”.

(14) Line 97. “ Individual Pb and Cd stress was individually applied” has modified to “Pb and Cd stress were individually applied”.

(15) Line 102~103. “Five milliliters of acetone were added to 0.5 g of leaf tissue, which then was incubated in darkness (4 �C for 72 h) until the color completely disappeared from the leaves.” Has modified to “Five millilitres of acetone was added to 0.5 g of leaf tissue, which was incubated in darkness (4 �C for 72 h) until the colour completely disappeared from the leaves.”

(16) Line 114~118. “The products of chlorophyll synthesis and degradation were determined via an enzyme-linked immunosorbent assay (ELISA) kit” has modified to “The activities of Chlase, MDCase, δ-aminolevulinic acid (δ-ALA), porphobilinogen (PBG) and uroporphyrinogen (Urogen Ⅲ) were measured by using a Chlase assay kit (LE-B044, 96T), MDCase assay kit (LE-B059, 96T), δ-ALA assay kit (LE-06543, 96T), PBG assay kit (LE-B255, 96T) and Urogen Ⅲ assay kit (LE-B254, 96T), respectively”.

(17) Line 118~119. “The ELISA kit involves a one-step sandwich enzyme-linked immunosorbent assay with double antibodies” has modified to “The enzyme-linked immunosorbent assay kit (ELISA) produced by Hefei Laier Bioengineering Institute was implemented according to the manufacturer’s instructions”.

(18) Line 122. “labeled” has modified to “labelled”. “were” was modified to “was”.

(19) Line 128. “Measurements of gas-exchange” has modified to “Gas-exchange measurements”

(20) Line 130. “for measuring” has modified to “to measure”.

(21) Line 134. “Plants metal content analysis ” has modified to “Metal content analysis of plants”.

(22) Line 158. “pigments concentration” has modified to “pigment concentrations”.

(23) Line 273-274. “These pigments provide the ability of plants to synthesize their own substances” has modified to “These pigments guarantee that plants are able to synthesize their own substances”.

(24) Line 274~276. “The results showed that the contents of chlorophyll a, chlorophyll b and total chlorophyll increased when the Pb concentration was 400 kg·mg-1. At high Cd concentrations, chlorophyll b did not significantly differ from that in the control treatment” has modified to “With an increase in the concentration of single Pb and combined stress, Chl-a, Chl-b and total chlorophyll showed increasing trends, and the concentrations were higher than those of the control group”.

(25) Line 291~300. “Under individual and combined stresses, chlorophyllase did not exhibit high activity , and the chlorophyll content was also high when the Pb and Cd concentrations were relatively high, both of which are related to the production of antioxidants.” has modified to “Chlase and MDCase play key roles in the first and second steps of the chlorophyll decomposition process, respectively, showing high activity at a Pb concentration of 400 mg·kg-1, Cd concentrations of 10 mg·kg-1 and 20 mg·kg-1 and combined stress concentrations of 200 + 1 mg·kg-1. Under stress due to high concentrations of Pb and Cd (1000 mg·kg-1 and 30 mg·kg-1), the activity of demerged chelatase, which plays a role in the second step of decomposition, was higher than that in the control group. This indicated that the degree of chlorophyll decomposition did not increase significantly under the stress of low heavy metal concentrations. Therefore, the increase in the chlorophyll concentration is related to the lower degree of chlorophyll decomposition (Kraj 2015). This also may be related to the enhancement of other resistance mechanisms in plants.”

(26) Line 318~320. “However, with the exception of that at 800 mg·kg-1Pb, the uroporphyrinogen III content did not significantly differ in response to individual Pb and combined stresses” ha smodified to “Under single Pb treatment, the concentration of uroporphyrinogen III was higher than that of the control group except for 800 mg·kg-1”. 

(27) Line 333~336. “In the present study, the intercellular CO2 concentration increased with increasing Pb and Cd concentrations, which may be due to the reduced photosynthetic rate and the long-term CO2 accumulation in the leaves; thus, stomatal limitation is not the main factor affecting the photosynthesis of D. involucrate” has modified to “In the present study, the intercellular CO2 concentration increased, and the transpiration rate increased first and then decreased with increasing Cd concentrations, which indicated that stomatal limitation is not the main factor affecting the photosynthesis of D. involucrate”.

(28) Line 364~367. “This study is the first to report the responses of D. involucrata photosynthetic pigments, chlorophyll synthesis and degradation products and gas-exchange parameters to metal stress, as well 

as the accumulation and translocation of those metals in different tissues. It was found that the tolerance of D. involucrata to high concentration of Pb and Cd was restored. Almost all of the Pb 

and Cd was retained in the roots of D. involucrata, which reduced their toxicity in the stems and leaves. Moreover, there was low translocation of these metals to the stems and leaves. The photosynthetic pigments were relatively little affected by the heavy metals” has modified to “The photosynthesis characteristics of D. involucrata in response to heavy metal stress and the Pb and Cd accumulation and translocation ability in different tissues were comprehensively reported for the first time. Photosynthetic pigments were slightly inhibited by Pb and Cd, and the synthesis of chlorophyll by D. involucrata was less affected by high concentrations of Pb and Cd.”

(29) Line 369~371. “ The gas-exchange parameters are more sensitive to heavy metal stress” has modified to “Gas-exchange parameters were more sensitive to heavy metal stress, and the net photosynthetic rate decreased with increased heavy metal concentrations”.

2. Lines 10-11: “We used one-way ANOVA to analyze the photosynthesis, accumulation and 

translocation in D. involucrate”. What accumulation and translocation? Need to be specific.

Reply:

 We thank the reviewer for the valuable comments. We have explained the accumulation and translocation.

(1) Line 9~11. “We used one-way ANOVA to analyse the photosynthesis, accumulation and translocation in D. involucrate” was modified to “We used ANOVA to analyse the photosynthesis of D. involucrata and the ability of lead and cadmium to enrich and migrate in roots, stems and leaves”.

3. What do you mean “accumulation ability” in line 13, BCF or TF?

Reply:

 We thank the reviewer for the valuable comments. We have explained “accumulation ability”.

(1)Line 11~12. “Accumulation ability” was explained as “accumulation factors”. Under individual and combined Pb and Cd stress, the accumulation factors in the roots were greater than 1, which was significantly greater than those in the stems and leaves (P<0.05)

4. Line 15: roots of plant always have higher enrichment for HMs than stems and leaves.

Reply:

We thank the reviewer for the valuable comments. We have modified this sentence.

(1) Line 14~15. We have modified “It has been found that the enrichment ability of roots to Pb and Cd is significantly higher than that of stems and leaves” to “The Pb and Cd enrichment ability of D. involucrata roots was significantly higher than that of stems and leaves”.

5. Line 17: “increased under high concentrations stresses.”, high conc. of what stress? Need to be specified.

Reply:

We thank the reviewer for the valuable comments. We have added some information.

(1)Line 16~17. “The Mg-dechelatase (MDCase) activities of chlorophyll degradation products increased under stress due to high concentrations” was modified to “The Mg-dechelatase (MDCase) activities of chlorophyll degradation products increased under stress due to high concentrations of Pb and Cd ”.

6. Line 18, “tow” should be “two”

Reply:

We thank the reviewer for the valuable comments.

(1) Line 18. “tow” was modified to “two”.

7. Line 22: “Pb and Cd in the range of 400+5~800+20 mg kg-1”, what is the symbol “+” stands for?

Reply:

We thank the reviewer for the valuable comments. We have explained the symbolic meaning of “+”.

(1) Line 21~22. The symbolic meaning of “+” is that it indicates the combined stress of Pb and Cd,

with the concentrations of Pb and Cd before and after the “+” respectively. We have modified the sentence to “Under combined stress, concentrations of Pb and Cd in the range of 400~800 mg·kg-1 and 5~20 mg·kg-1”.

8. Line 25: “The photosynthesis of D. involucrata has strong tolerance to higher concentrations of heavy metals.”, what do the authors mean "photosynthesis has strong tolerance"?

Reply:

We thank the reviewer for the valuable comments. We want to express that the photosynthesis tolerance of D. involucrata is stronger than the stress of heavy metals. We have deleted this statement and elaborated upon it in the conclusion.

(1) Line 366~371. “Photosynthetic pigments were slightly inhibited by Pb and Cd, and the synthesis of chlorophyll by D. involucrata was less affected by high concentrations of Pb and Cd. Chlorophyll synthesis products increased with increasing concentrations of Pb and Cd, and at the same time, the degradation products decreased. Gas-exchange parameters were more sensitive to heavy metal stress, and the net photosynthetic rate decreased with increased heavy metal concentrations”.

(2 Line 25. We have de deleted the sentence of “The photosynthesis of D. involucrata has strong tolerance to higher concentrations of heavy metals”. 

(3) Line 25~26. “In addition, the root of D. involucrata has a strong absorption and fixation effect on heavy metals, thereby reducing metal toxicity” has modified to “In addition, the root of D. involucrata had a strong absorption and fixation effect on heavy metals, thereby reducing metal toxicity and improving the tolerance of D. involucrata to heavy metals.”

9. Lines 33-34: “Lead (Pb) and cadmium (Cd) are nondegradable, are easily translocated, and are long-lived in the soil”, The translocation ability of these two metals varied with plant species, can be easier in some plants but harder in some others. To my point of view, Pb is always hard to be translocated in most plants.

Reply:

We thank the reviewer for the valuable comments. According to the reviewer's opinion, we made a modification.

(1) Line 32~33. “Lead (Pb) and cadmium (Cd) are nondegradable, are easily translocated, and are long-lived in the soil” was modified to “Lead (Pb) and cadmium (Cd) are nondegradable, long-lived and exhibit strong toxicity in the soil”.

10. Lines 38-39: “The absorption capacity for heavy metals of different plants parts is as follows: roots > stems > leaves (Arena et al., 2017; zhu et al., 2007; Keller et al., 2003). ”. The absorption capacity of HMs in plants depends on the plant species and metal types. not for all the plants, hyperaccumulator sometimes accumulate more HMs in the aboveground parts.

Reply:

We thank the reviewer for the valuable comments. According to the reviewer's opinion, we made a modification.

(1) Line 37~38. “The absorption capacity for heavy metals of different plants parts is as follows: roots > stems > leaves” was modified to “For most plant species, roots represent a barrier for metals. Therefore, the concentration of heavy metals in roots is usually higher than that of stems and leaves”.

11. Lines 96-97: The conc. of Pb and Cd used in the present study are too high, the author should state the reason in the manuscript why they choose such high levels of HMs and prove the levels used were environmental realistic.

Reply:

We thank the reviewer for the valuable comments. We have included the following explanations for the problems mentioned above. According to Soil Environmental Quality Standards, GB15618-1995, China, we know that soil environmental quality standard values at three levels are Pb≤500 mg·kg-1, Cd≤1 mg·kg-1. In China, the highest levels of Pb and Cd pollution can reach 1143 mg·kg-1 and 228 mg·kg-1, respectively, according to Gao et al., 2011. Moreover, We want to simulate the effects of mild, moderate and severe HMs pollution.

(1) Line 90. We have added this sentence “the three levels of soil environmental quality standard values are Pb≤500 mg·kg-1 and Cd≤1 mg·kg-1”

(2) Line 91~92. We have added this sentence “In China, the highest levels of Pb and Cd pollution can reach 1143 mg·kg-1 and 228 mg·kg-1, respectively, according to Gao et al., 2011.”

(3) Line 92~95. “the individual index method for assessing soil heavy metal pollution for simulating mild, moderate and severe heavy metal pollution, we adopted an orthogonal experimental design method in the establishment of 16 concentration gradients with three replicates per treatment.Pb(NO3)2 and CdCl2·2.5H2O were used to generate different concentrations of solutions” has modified to “We adopted an orthogonal experimental design method to establish 16 concentration gradients to simulate the effects of mild, moderate and severe pollution of heavy metals on the photosynthesis of D. involucrata. Pb(NO3)2 and CdCl2·2.5H2O were used to generate different concentrations of solutions”

12. Line 100: Do you have any reference to support the using of acetone to extract the leaf chlorophylls? 

Reply:

We thank the reviewer for the valuable comments. We added the references.

(1) We have used acetone to extract chlorophyll according to references such as Lichtenthaler et al.,1987; Tauqeer et al., 2016 and Porra, 2002; Zhong, et al., 2017).

13. Line 105: which reference exactly is the one that your calculation based on?

Reply:

We thank the reviewer for the valuable comment. We added the reference.

(1) Lichtenthaler et al. (1987)

(2) Tauqeer et al. (2016)

(3) Porra (2002)

(4) Zhong, et al. (2017)

14. Lines 129-134: What is the determination limit? Did you use any certified reference materials to check the recovery rate of your digestion process, and what is the recovery rate of these metals?

Reply:

We thank the reviewer for the valuable comments. Here is our explanation.

(1) We rely on the Ecological Experimental Station of Red Soil, Chinese Academy of Sciences to determine the heavy metal content in roots, stems and leaves of D. involucrata in this experiment. The detection methods used are internationally recognized.

(2) In this experiment, we only need the content data of heavy metals in various tissues, and calculate the accumulation factors and translocation factors based on these data.

15. For the calculation of BCF and TF, I believe there are more suitable reference can be cited other than Wu et al. (2012)

Reply:

We thank the reviewer for the valuable comment. 

(1) Line 139. We replaced the previous reference with Bahri et al., 2015.

16. Line 151: “were greatest” should be “were the greatest”

Reply:

We thank the reviewer for the valuable comment.

(1) Line 160. “were greatest” was modified to “were the greatest”.

17. Figure 1: The abbreviations, Chlase and MDCase in Figure 1, are not defined previously.

Reply:

We thank the reviewer for the valuable comment. We have defined Chlorophyllase (Chlase) and Mg-dechelatase (MDCase). 

(1) Line 65~66. We have added the sentences of “Zhou et al. (2011) reported that chlorophyllase (Chlase) and Mg-dechelatase (MDCase) can cause the decomposition of chlorophyll ”.

18. The color of different columns in Fig.1: please consider to change the columns for Pb treatment to white. Same problems also shown for Fig.2 and fig.6.

Reply:

We thank the reviewer for the valuable comments. We have changed the columns for Pb treatment to white in Fig. 1, Fig. 2 and Fig. 6.

(1) Previous Fig. 1 corresponds to current Fig. 2.

Fig. 2 Effects of different concentrations of Pb and Cd on the activities of chlorophyllase and Mg-dechelatase

(2) Previous Fig. 2 corresponds to current Fig. 3.

Fig. 3 Effects of different concentrations of Pb and Cd on the contents of δ-aminolevulinic acid porphobilinogen, and uroporphyrinogen III

(3) Previous Fig. 6 corresponds to current Fig. 7.

Fig. 7 (a), (b) BCFs of Pb and BCFs of Cd in different plant parts under individual stresses. (c), (d) BCFs of Pb and BCFs of Cd in different plant parts under combined stress

19. For the data shown in Figures 1-5: the data of the different treatment does not seem to be significantly different from each other. You mentioned a Duncan test at 5% probabilities was used for comparison. Duncan's test is not a good choice when it comes to comparing treatment means. Authors should use an appropriate statistical test for this purpose and modify the content of the manuscript.

Reply:

We thank the reviewer for the valuable comments. We have thought about it carefully, and we use Tukey's test to analyse all of the data.

(1) Line 150~155. “The experimental results of this study are presented as the mean of three replicates. Differences among treatments were analysed by one-way analysis of variance (ANOVA), and the significance of interactions between Pb and Cd was analysed by use of two-way ANOVA. The least significant difference (Tukey’s test) was applied to determine the significance between different treatments, and the critical value for statistical significance was P < 0.05. All statistical analyses were carried out using SPSS 23.0 (SPSS, Chicago, USA).”

20. Some data in Figs. 4 do not seem to be significantly different. Take Figure 4a as an example, I don’t believe the data B4 and B5 are significantly different. The author should double-check the statistical results of the data in all the figures.

Reply:

We thank the reviewer for the valuable comments. We have modified the data processing method.

(1) Differences among treatments were analysed by one-way analysis of variance (ANOVA), and the significance of interactions between Pb and Cd was analysed by two-way ANOVA. Tukey’s test and double-check was applied to determine the significance between different treatments.

21. Table 2: Define “transfer coefficients”. Please include the calculation details in the materials and methods part. 

Reply:

We thank the reviewer for the valuable comments. 

(1) Line . “transfer coefficients” is “translocation factor”. We have modified “transfer coefficients” to “TF”.

22.Table 1 and Table 2: Using superscript lower-case letters to indicate the significant differences.

Reply:

We thank the reviewer for the valuable comments. We changed Table 1 to columns to better present the changes in chlorophyll.

Fig. 1 Analysis of the differences in photosynthetic pigments of D. involucrata under different concentrations of Pb and Cd 

23. Line 342: “This study is the first to report the responses of..” the first what? need to rephrase.

Reply:

We thank the reviewer for the valuable comments. We have rephrased “the first”.

(1) Line 364~366. We have modified “This study is the first to report the responses of D. involucrata

photosynthetic pigments, chlorophyll synthesis and degradation products and gas-exchange parameters to metal stress, as well as the accumulation and translocation of those metals in different tissues.” to “The photosynthesis characteristics of D. involucrata in response to heavy metal stress and the Pb and Cd accumulation and translocation ability in different tissues were comprehensively reported for the first time.”

24. Line 345: “It was found that the tolerance of D. involucrata to high concentration of Pb and Cd was restored.”, what do you mean restored? restored by what? Need to specify.

Reply:

We thank the reviewer for the valuable comments. We have specified the sentence.

(1) Line 366. We have modified “It was found that the tolerance of D. involucrata to high concentration of Pb and Cd was restored” to “Photosynthetic pigments were slightly inhibited by Pb and Cd”.

25. Line 348: “The photosynthetic pigments were relatively little affected by the heavy metals.”， please rephrase this sentence.

Reply:

We thank the reviewer for the valuable comments. We have rephrased this sentence.

(1) Line 367~368. We have modified “The photosynthetic pigments were relatively little affected by the heavy metals.” to “the synthesis of chlorophyll by D. involucrata was less affected by high concentrations of Pb and Cd”.

Reviewer #2: This is a nice paper reporting results about the "Response of photosynthesis to different concentrations of heavy metals in Davidia involucrata" - I would try to specify it more clearly and loudly in both the abstract and the conclusion, which are pretty weak now, could be much more catchy.

1.The study seems to be run using sounds laboratory techniques, but not robust statistical methods and data interpretation. Particularly, since the Authors are testing 2 different heavy metals (Pb and Cd) a 2-way ANOVA is needed.

Reply:

We thank the reviewer for the valuable comments. We considered the reviewer's suggestions and adopted a more appropriate analysis method. 

(1) Line 150~155. Differences among treatments were analysed by one-way analysis of variance (ANOVA), and the significance of interactions between Pb and Cd was analysed by two-way ANOVA. The least significant difference (Tukey’s test) was applied to determine the significance between different treatments, and the critical value for statistical significance was P < 0.05. All statistical analyses were carried out using SPSS 23.0 (SPSS, Chicago, USA).

2. I like this manuscript because so straightforward and a better statistical analysis of the results would increase the value. Also, since this plants is endemic from China, maybe adding a photo of the species would be a good idea.

Reply:

We thank the reviewer for the valuable comments. We have attached a picture of Davidia involucrata at the end.

(1) D. involucrata seedlings used in the experiments

(2) D. involucrata

Reviewer #3: The study deals with important topic of heavy metal effects in plants. The authors tried to address the effects of different concentrations of Pb and Cd on photosynthesis in a very specific wood plant Davidia involucrata, an endangered woody species. The authors focused on allocation of heavy metals in plants and effects of different concentrations on chlorophyll content, photosynthesis, stomatal activity as well as selected biochemical analyses. Overall, the manuscript is written well and presented clearly. However, I have some serious doubts related to some methodical aspects and interpretation of the data. Especially, the lead (Pb) was applied in form of nitrogen containing molecule Pb(NO)2. Despite the content of nitrogen in this molecule is not very high (app. 10%), considering a very high doses applied, the plants got significant amount of nitrogen, which could lead to significant fertilizing effects. This well explains increase of chlorophyll content or trend of photosynthetic characteristics observed in this study. The effect of additional nitrogen was not considered in methodic nor in interpretation of the data. Hence, there is a high risk that the recent version of the study contains significant artifacts and, therefore, it is not suitable for publication in present form. I strongly encourage the authors re-consider the results and use only the part in which there is no doubt regarding the methods. In my point of view, the manuscript is not appropriate for publishing and should be rejected.

Reply:

We thank the reviewer for the valuable comments. We explained the comments of the reviewers as follows.

(1) Pb(NO3)2 used in this experiment is a commonly used heavy metal addition reagent. (Bezerril Fontenele et al., 2017; Zhong, et al., 2017)

Bezerril Fontenele N M , Otoch M D L O , Gomes-Rochette, Neuza Félix, et al. Effect of lead on physiological and antioxidant responses in two, Vigna unguiculata, cultivars differing in Pb-accumulation[J]. Chemosphere, 2017, 176:397-404.

Zhong B, Chen J, Shafi M, et al. Effect of lead (Pb) on antioxidation system and accumulation ability of Moso bamboo (Phyllostachys pubescens)[J]. Ecotoxicology and Environmental Safety, 2017, 138: 71-77.

(2) The small amount of nitrate present in this study has no significant effect on the results of this study. Many studies have shown that heavy metals will directly damage the plant tissue and cell structure, and the absorption of nutrients will be limited in the growth process (Rizwan et al., 2018; Shi et al., 2014; Azzarello et al., 2012；Sun et al., 2018; Pilipović et al., 2019). In addition, the experimental seedlings are three years old, and they are not in the stage with a high demand for nutrients. Therefore, the heavy metal reagents added in this study have little impact on the experimental results.

Rizwan M, Ali S, Abbas T, et al. Residual effects of biochar on growth, photosynthesis and cadmium uptake in rice (Oryza sativa, L.) under Cd stress with different water conditions[J]. Journal of Environmental Management, 2018, 206:676-683.

Shi G, Xia S, Ye J, et al. PEG-simulated drought stress decreases cadmium accumulation in castor bean by altering root morphology[J]. Environmental and Experimental Botany, 2014, 111:127-134.

Sun X, Xu Y, Zhang Q, et al. Combined effect of water inundation and heavy metals on the photosynthesis and physiology of, Spartina alterniflora[J]. Ecotoxicology and Environmental Safety, 2018, 153:248-258.

Pilipović A, Zalesny Jr R S, Rončević S, et al. Growth, physiology, and phytoextraction potential of poplar and willow established in soils amended with heavy-metal contaminated, dredged river sediments[J]. Journal of environmental management, 2019, 239: 352-365.

---

## [Decision Letter · Decision Letter 1]

21 Jan 2020

Response of photosynthesis to different concentrations of heavy metals in Davidia involucrata

PONE-D-19-28304R1

Dear Dr. Yang,

We are pleased to inform you that your manuscript has been judged scientifically suitable for publication and will be formally accepted for publication once it complies with all outstanding technical requirements.

With kind regards,

Mayank Gururani

Academic Editor

PLOS ONE

Additional Editor Comments (optional):

Reviewers' comments:

Reviewer's Responses to Questions

**Comments to the Author**

1. If the authors have adequately addressed your comments raised in a previous round of review and you feel that this manuscript is now acceptable for publication, you may indicate that here to bypass the “Comments to the Author” section, enter your conflict of interest statement in the “Confidential to Editor” section, and submit your "Accept" recommendation.

Reviewer #1: All comments have been addressed

Reviewer #2: All comments have been addressed

2. Is the manuscript technically sound, and do the data support the conclusions?

Reviewer #1: Yes

Reviewer #2: Yes

3. Has the statistical analysis been performed appropriately and rigorously? 

Reviewer #1: Yes

Reviewer #2: Yes

4. Have the authors made all data underlying the findings in their manuscript fully available?

Reviewer #1: Yes

Reviewer #2: Yes

5. Is the manuscript presented in an intelligible fashion and written in standard English?

Reviewer #1: Yes

Reviewer #2: Yes

6. Review Comments to the Author

Reviewer #1: Most of my comments has been addressed properly, although I am still not fully convinced by some of them.

Reviewer #2: The Authors addressed all my comments. Statical analysis has been improved and the manuscript is now ready for publication.

7. PLOS authors have the option to publish the peer review history of their article (what does this mean?). If published, this will include your full peer review and any attached files.

Reviewer #1: No

Reviewer #2: Yes: Lorenzo Rossi

---

## [Editor Report · Acceptance letter]

7 Feb 2020

PONE-D-19-28304R1 

 Response of photosynthesis to different concentrations of heavy metals in *Davidia involucrata*

Dear Dr. Yang:

I am pleased to inform you that your manuscript has been deemed suitable for publication in PLOS ONE. Congratulations! Your manuscript is now with our production department. 

With kind regards,

on behalf of

Dr. Mayank Gururani 

Academic Editor

PLOS ONE